

# Technical note: Analysis of observation uncertainty for flood assimilation and forecasting

Joanne A. Waller[1], Javier García-Pintado[1,2], David C. Mason[3], Sarah L. Dance[1], and Nancy K. Nichols[1]

[1]School of Mathematical, Physical and Computational Sciences, University of Reading, UK
[2]MARUM – Center for Marine Environmental Sciences and Department of Geosciences, University of Bremen, Germany
[3]School of Archaeology, Geography and Environmental Science, University of Reading, UK

*Correspondence to:* Joanne A. Waller (j.a.waller@reading.ac.uk)

**Abstract.** The assimilation of satellite-based water level observations (WLOs) into 2D hydrodynamic models can keep flood forecasts on track or be used for reanalysis to obtain improved assessments of previous flood footprints. In either case, satellites provide spatially dense observation fields, but with spatially correlated errors. To date, assimilation methods in flood forecasting either incorrectly neglect the spatial correlation in the observation errors or, in the best of cases, deal with it by thinning methods. These thinning methods result in a sparse set of observations whose error correlations are assumed to be negligible. Here, with a case study, we show that the assimilation diagnostics that make use of statistical averages of observation-minus-background and observation-minus-analysis residuals are useful to estimate error correlations in WLOs. The estimated correlations do not behave as expected; however, analysis shows that the diagnostic can also be used to highlight anomalous observation datasets. Accurate estimates of the observation error statistics can be used to support quality control protocols and provide insight into which observations it is most beneficial to assimilate. Furthermore, the understanding gained in this paper will contribute towards the correct assimilation of denser datasets.

## 1 Introduction

In data assimilation (DA), observations are combined with numerical model output, known as the background, to provide an accurate description of the current state, known as the analysis. In DA the contributions from the background and observations are weighted according to their relative uncertainty. In the assimilation, both the instrument error and representation error contribute to the observation error (Janjić et al., 2017), which may be correlated and state dependent (Waller et al., 2014; Hodyss and Nichols, 2015). In DA, observation error statistics are typically assumed to be uncorrelated. The data density is reduced in order to satisfy this assumption (Lorenc, 1981). Having adequate estimates of these uncertainties is crucial in order to obtain an accurate analysis. In numerical weather prediction (NWP) DA systems the use of full observation error correlation matrices has lead to the inclusion of additional observation information content (Stewart et al., 2008). This results in a more accurate analysis and improvements in objectively measured forecast skill (Weston et al., 2014; Bormann et al., 2016). Furthermore, an understanding of the observation uncertainties can provide insight into which observations are most useful to assimilate.





The development of DA systems has largely been driven by its use in NWP, but the methodologies are applicable to any system that can be modelled and observed. There have been recent advances in real-time 2D hydrodynamic modeling and the acquisition and processing of relevant remote sensing observations (earth observations, EOs) (Raclot, 2006; Andreadis et al., 2007; Schumann et al., 2007, 2011; Mason et al., 2010a, 2012, 2014). Consequently, several studies have shown the benefit

of applying DA to operational flood forecasting (Durand et al., 2008, 2014; Montanari et al., 2009; Roux and Dartus, 2008; Neal et al., 2009; Matgen et al., 2010; Mason et al., 2010b; Giustarini et al., 2011; García-Pintado et al., 2013, 2015). Grimaldi et al. (2016) review the potential of EOs for inundation mapping and water level estimation and their use for calibration, validation and constraint of real-time hydraulic flood forecasting models. A predominant EO technique to obtain WLOs is Synthetic Aperture Radar (SAR). SAR provides high-resolution observations of radar backscatter which, after processing,

serve to delineate the flood extent. Then, the intersection of the flood extent with a high-resolution LiDAR Digital Terrain Model is used to obtain the WLOs.

A common characteristic for these EOs, is that they need to be subjected to strict quality control (QC) procedures, if they are to be unbiased. The QC, for example, may reject observations in vegetated areas or in other conditions depending on its application. As a result the observed flood extent is discontinuous in space. Nevertheless, these discontinuous observations may

be rather dense and the errors in the observations may be highly correlated. A direct assimilation of this dense dataset would lead to an analysis biased towards the observations and, for covariance-evolving methods (e.g., ensemble Kalman filters), an over reduced posterior covariance and unstable long-term forecast/assimilation cycles. Thus, to avoid dealing with the spatial correlation in the assimilation, the current approach is to further thin the data, as is standard in other assimilation applications, typically retaining approximately 1% of the pre-thinned observations. The result is that the dataset to be assimilated is a sparse

field of clustered observations.

Irrespective of the processing method, the full correlated observation error statistics are unknown. In the field of hydrological forecasting, one scenario that would benefit from improved understanding of the observation uncertainties is the assimilation of satellite-derived surface soil moisture (SSM) into catchment-scale rainfall-runoff models (Cenci et al., 2016; Mason et al., 2016). Another, which is receiving great attention, is the assimilation of the satellite-derived water level observations (WLOs)

for either operational flood forecast or hindcast analyses (Mason et al., 2010a; García-Pintado et al., 2013). A more detailed understanding of the observation uncertainties would be highly useful as they can inform the thinning strategy and suggest which observations may benefit the assimilation most (Fowler et al., 2017). Additionally, understanding the error statistics may permit more observations to be included in the assimilation, which should allow the information from dense observation sets to be fully exploited. There is a clear potential to improve the flood forecast if all the SAR WLOs could be assimilated in an

appropriate way.

Observation uncertainties cannot be computed directly, instead they must be statistically estimated. Desroziers et al. (2005) provide a diagnostic to estimate observation uncertainties using the statistical average of observation-minus-background and observation-minus-analysis residuals. The diagnostic has been applied to operational NWP settings to estimate observation uncertainties with good results (Weston et al., 2014; Waller et al., 2016a, c; Bormann et al., 2016; Cordoba et al., 2017). In this

manuscript we use the diagnostic of Desroziers et al. (2005), described in Section 2, to estimate the observation error statistics





for SAR WLOs that are assimilated using a Local Ensemble Transform Kalman Filter (LETKF) into the LISFLOOD-FP 2D hydrodynamic model. For this study, we use a sequence of real SAR overpasses in a flood event that occurred in November 2012 in SW England. A description of the SAR WLOs and experimental design are given in Section 3.

Results are discussed in Section 4. First, we estimate average WLO error statistics across the entire domain for the duration of the flood event. It will be seen later that these globally estimated error statistics show an anomalous pattern. Thus, we then consider if these error statistics vary across the domain or for different phases of the flood. From the results we infer that the anomalous pattern is related not related to the distribution of observations over the domain, but to observations during the later stages of the flood. Importantly, we show that the diagnostic of Desroziers et al. (2005) can be used to identify anomalous observation datasets that are not suitable for assimilation.

## 2 The diagnostic of Desroziers et al. (2005)

Data assimilation is a technique used to provide an analysis, $\mathbf{x}^a \in \mathbb{R}^{N^m}$, the best estimate of the current state of a dynamical system. The analysis is determined by combining the background $\mathbf{x}^b \in \mathbb{R}^{N^m}$, a model prediction, with observations, $\mathbf{y} \in \mathbb{R}^{N^p}$, weighted by their respective error statistics. Here the dimensions of the observation and model state vectors are denoted by $N^p$ and $N^m$, respectively. To combine the information from the observations and background it is necessary to project the background into observation space using the observation operator, $\mathcal{H} : \mathbb{R}^{N^p} \to \mathbb{R}^{N^m}$ which may be non-linear. The analysis can be used to initialise a forecast which in turn provides a background for the next assimilation.

In Desroziers et al. (2005) the analysis is calculated using

$$
\begin{aligned}
\mathbf{x}^a &= \mathbf{x}^b + \mathbf{B}\mathbf{H}^T(\mathbf{H}\mathbf{B}\mathbf{H}^T + \mathbf{R})^{-1}(\mathbf{y} - \mathcal{H}(\mathbf{x}^b)), \\
&= \mathbf{x}^b + \mathbf{K}(\mathbf{y} - \mathcal{H}(\mathbf{x}^b)),
\end{aligned}
\tag{1}
$$

where $\mathbf{R} \in \mathbb{R}^{N^p \times N^p}$ and $\mathbf{B} \in \mathbb{R}^{N^m \times N^m}$ are the observation and background error covariance matrices, $\mathbf{K}$ is the Kalman gain matrix and $\mathbf{H}$ is the linearised observation operator, linearised about the background state.

The observation error covariance matrix can be estimated using the observation-minus-background, $\mathbf{d}_b^o = \mathbf{y} - \mathcal{H}(\mathbf{x}^b)$, and observation-minus-analysis, $\mathbf{d}_a^o = \mathbf{y} - \mathcal{H}(\mathbf{x}^a)$, residuals (Desroziers et al., 2005). Assuming that the observation and background errors are mutually uncorrelated, the statistical expectation of the product of the analysis and background residuals results in

$$
E[\mathbf{d}_a^o \mathbf{d}_b^{oT}] \approx \mathbf{R}.
\tag{2}
$$

As the resulting matrix is estimated statistically it will not be symmetric. Therefore, it must be symmetrised before it can be used in a data assimilation scheme .

The form of the diagnostic in Eq. (2) is not suitable to calculate observation error statistics when each assimilation cycle uses different observations. Instead components of the background and analysis residuals must be paired and binned, with the binning dependent on the type of correlation being estimated. For example, when calculating spatial correlations the bins may



depend on the distance between observations, where as for temporal correlations the bins would depend on the time between observations. For each bin, $\beta$, the covariance, $cov(\beta)$ is then computed individually using,

$$cov(\beta) = \frac{1}{N^\beta} \sum_{k=1}^{N^\beta} \left( \mathbf{d}_i^{oa} \mathbf{d}_j^{ob} \right)_k - \frac{1}{N^\beta} \sum_{k=1}^{N^\beta} (\mathbf{d}_i^{oa})_k \frac{1}{N^\beta} \sum_{k=1}^{N^\beta} (\mathbf{d}_j^{ob})_k, \tag{3}$$

where $\left( \mathbf{d}_i^{oa} \mathbf{d}_j^{ob} \right)_k$ is the $k^{th}$ pair of elements of $\mathbf{d}_a^o$ and $\mathbf{d}_b^o$ in bin $\beta$, and $N^\beta$ is the number of residual pairs in bin $\beta$. The

5 second term of equation (3) ensures that the computation of the observation error statistics is not affected by bias (Waller et al., 2016a).

The diagnostic in Eqs. (2) and (3) only gives a correct estimate of the observation error uncertainties if the error statistics used in the assimilation are exact. Even if the assumed statistics are not exact the diagnostic can still provide useful information about the true observation error statistics (Waller et al., 2016b; Ménard, 2016). Further limitations include the use of an ergodic

assumption in order to obtain sufficient samples (Todling, 2015) and the assumption that the observation operator is linear (Terasaki and Miyoshi, 2014).

One further issue is that the standard diagnostic is derived assuming that the analysis is calculated using minimum variance linear statistical estimation. If local ensemble DA is used to determine the analysis, the diagnostic does not result in a correct estimate of the observation uncertainties. However, by using a modified version of the diagnostic some of the observation error

statistics may be estimated. It is possible to estimate the error correlations between two observations if the observation operator that determines the model equivalent of observation $\mathbf{y}_i$ acts only on states that have been updated using the observation $\mathbf{y}_j$ (Waller et al., 2017). Since we use a LETKF assimilation scheme in this study, we must take this into account when estimating observation error statistics for the WLOs.

## 3  Experimental Design

This study makes use of the observation, model and assimilation system described in García-Pintado et al. (2015). We direct the reader to this reference, and references therein, for a detailed description of the derivation of WLOs and the assimilation design. Here we provide a description of the data used specifically in this study.

We estimate observation uncertainties for observations from a real flood event that occurred in the South West United Kingdom on a $30.6km \times 49.8km$ (1 524 km$^2$) area of the lower Severn and Avon rivers in November 2012. The WLOs were

25 extracted from a sequence of seven satellite SAR observations (acquired by the COSMO-SkyMed constellation) using the method described in Mason et al. (2012). The WLOs are available daily for the period $27^{th}$ November to the $4^{th}$ December 2012 (with the exception of December $3^{rd}$). Observations on the first day illustrate the flood levels just before the flood peak in the Severn. On $30^{th}$ November the river went back in bank; however, a substantial amount of water remained on the floodplain (see Fig.2 in García-Pintado et al., 2015).

When presented with an unlikely observation, such an observation with a gross error measurement error or one which reflects a rare fluctuation in the dynamics of the system (Vanden-Eijnden and Weare, 2013), data assimilation techniques can lose accuracy. Thus, before being assimilated, the observations are subjected to several QC protocols according to the physical



characteristics of the terrain and land cover. The data are then thinned to a separation distance at which the observation errors are assumed uncorrelated. Typically the data is thinned to a separation distance of 250m. However, in this study a denser observation set (although still sparse) with thinning distance of 125 m is used, in which some spatial correlation should remain. The measured standard deviation for the WLOs is 59cm; this is calculated by fitting a plane by linear regression to the WLOs,

5 which we consider adequate in this case as the floodplain in the downstream observed areas is reasonably flat. The location of the observations is plotted in Fig. 1.

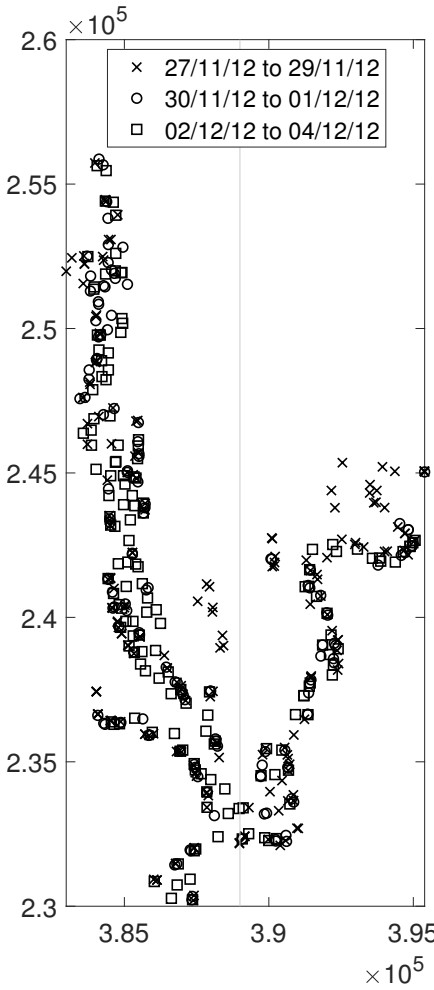

**Figure 1.** Position of SAR WLOs in flood model domain, OSGB 1936 British National Grid projection; coordinates in meters. Crosses: 27th, 28th 29th November, Circles: 30th November and 1st December, Squares: 2nd and 4th December. The line denotes the West/East domain split discussed in Section 4.2.





The observations are assimilated into a 75m resolution LISFLOOD-FP flood simulation model (Bates and Roo, 2000) using a LETKF (Hunt et al., 2007). Due to the formulation with the diagnostic described in Section 2, the localization in the LETKF is set in standard 2D Euclidean space rather than the physically based distance along the river channel described in García-Pintado et al. (2015), which would require a further adaptation of the diagnostic calculation. The localization radius is set using a compactly supported $5^{th}$ order piecewise rational function (Gaspari and Cohn, 1999, Eq. 4.10) with length scale 20km.

We apply the diagnostic of Desroziers et al. (2005) to the observation-minus-background and observation-minus-analysis residuals resulting from the flood assimilation. We first estimate average horizontal error covariances across the entire domain for the duration of the flood event. We then consider if these error statistics vary across the domain or for different phases of the flood. For all cases the observation error correlations are calculated at a 1km resolution. Due to issues with the diagnostic (discussed at the end of section 2), we do not consider any observation pairs with a separation distance greater than 19km. When evaluating the correlations we assume that they become insignificant when they drop below 0.2 (Liu and Rabier, 2002).

For this assimilation system we assume that the ensemble background error covariance matrix gives a reasonable estimate of the true background error statistics. The assumed standard deviation for the WLOs is 59cm; however, this does not account for the error of representation and, therefore, may be an underestimate of the true error standard deviation. As is typical for most DA systems, the observation errors are assumed uncorrelated. With these assumed error statistics the theoretical work of Waller et al. (2016b) suggests that the observation error statistics estimated using the diagnostic will have:

- An underestimated standard deviation.

- An underestimated correlation length-scale

Therefore when considering our results, we would expect the true standard deviations and length scales to be larger than those we estimate.

## 4 Results

### 4.1 Average observation error statistics

We first estimate average horizontal error covariances across the entire domain for the duration of the flood event. We plot in Fig. 2 the estimated correlation, along with the number of samples used, for the WLOs.

The estimated statistics give a standard deviation of 54cm, this is slightly lower that the assumed error standard deviation of 59cm. Following the theory of Waller et al. (2016b) we expect the estimated standard deviation to be an underestimate of the true observation error standard deviation and hence the results suggest that the assumed standard deviation is likely set at the correct level.

Our results show that the correlations become insignificant ($< 0.2$) at approximately 8km, but there is some unexpected behavior before 8km. The correlations drop smoothly between 0-4km then increase again up to 6km before dropping off. This behaviour is seen for a variety of different binning widths (not shown). We investigate the cause of this 'shoulder' in the



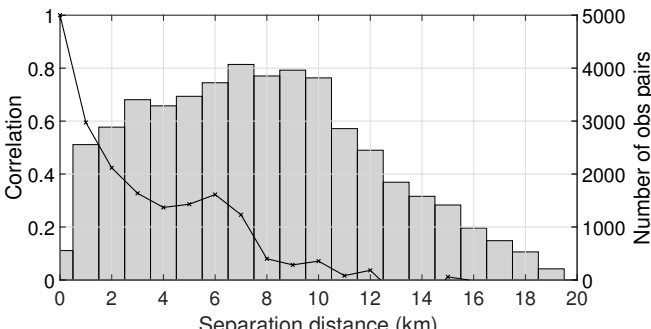

**Figure 2.** Estimated SAR WLO error correlations (black line) and number of samples (bars) used for the calculation. Estimated error standard deviation 54cm.

estimated correlations in Sections 4.2 and 4.3. In general we find that the correlation distance is much longer than the thinning distance of 125m which was chosen to try to ensure that the observation errors are uncorrelated. Furthermore, theoretical results of Waller et al. (2016b) suggest that, with this design of assimilation experiment, the correlation length scales will be underestimated.

5   **4.2   Correlations in different parts of the domain**

One possible cause of the shoulder in the correlations is the river structure. It is possible that observations on different tributaries of the river are resulting in the increase in correlations. To test this hypothesis we split the domain in two (as shown in Fig. 1); the western domain covering the river Severn and eastern domain covering the river Avon. We plot the estimated correlations, along with the number of samples used for the SAR WLOs, for the western part of the domain in Fig. 3 and for the eastern part

10  of the domain in Fig. 4. We note that there are fewer observations in the eastern domain, and therefore the results are subject to greater sampling error.

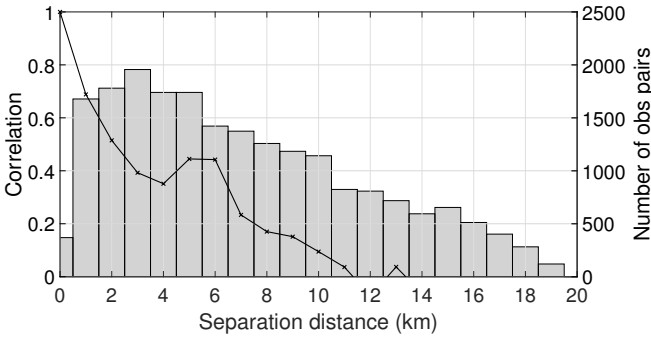

**Figure 3.** Estimated SAR WLO error correlations (black line) and number of samples (bars) used (bin width = 1km). West domain. Estimated error standard deviation 58cm.




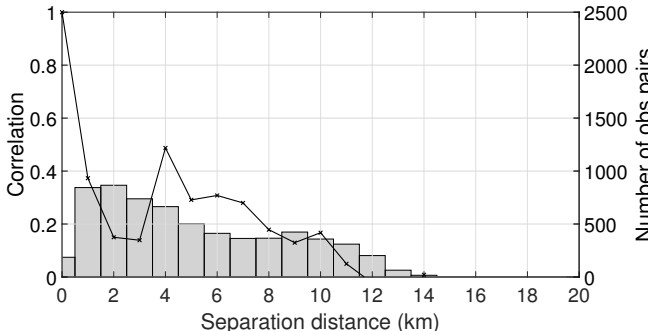

**Figure 4.** Estimated SAR WLO error correlations (black line) and number of samples (bars) used (bin width = 1km). East domain. Estimated error standard deviation 43cm.

From Fig.s 3 and 4 we see that the 'shoulder' in the correlations is still present in both parts of the domain. In the eastern domain it is very pronounced. This suggests that the cause of the increase in correlations between 4–6km is not observations on different tributaries of the river.

### 4.3 Correlations at different times

We next consider if the correlation structure changes over time. We plot in Fig.s 5, 6 and 7 the correlations calculated for the first three days, the second two days and the final two days respectively. We see that at the beginning of the flood period, the observations have similar standard deviations to those estimated for the entire flood event; however, the correlation length scale is short, approximately 2km. During the middle of the flood event the observation error standard deviation decreases and the correlation length scale increases slightly. For the final two days the river is back in bank; for this period the standard deviation is largest, as is the correlation length scale, which is approximately 8km. It is also in this final period where the 'shoulder' appears in the correlations.

In the recession stages of the flood, the days indicated in Fig. 7, the flow was back in bank. Thus an increasingly high proportion of the observations were in areas which remained flooded but were disconnected from the main river flow. For this same sequence of SAR overpasses García-Pintado et al. (2015) showed that the assimilation of the last three overpasses was still able to exploit the background ensemble covariances to pass some of the information from these WLOs to the main flow. However, two effects became evident: a) the assimilation increments to keep the forecast on track, despite being in the correct direction, were of a smaller magnitude (thus, not so effective) in these last stages, and b) the corrections to the flow in these last stages were gradually more short-lived. This was a result of the reduced information content in these WLOs regarding the inflow errors upstream, which in the end control the flood and flow evolution. Here the Desroziers et al. (2005) diagnostic has been able to identify a corresponding anomalous structure in the WLO errors at these last stages. The correlation structure showed in Fig. 7 indicates that apart from the longer correlation errors, which can be expected from the smoother flood dynamics at the end of the flood, an increase in the correlation appears at ∼ 6km. The increasing disconnection of the





WLOs in the flood plain from the main flow appears to be the cause for the shoulder in the estimated correlation structure. However, further work is required to determine why the 'shoulder' in the estimated correlation function appears at 6km.

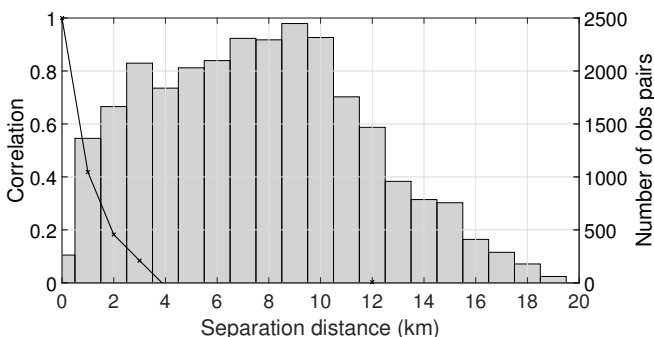

**Figure 5.** Estimated SAR WLO error correlations (black line) and number of samples (bars) used (bin width = 1km). 27th, 28th 29th November. Estimated error standard deviation 53cm.

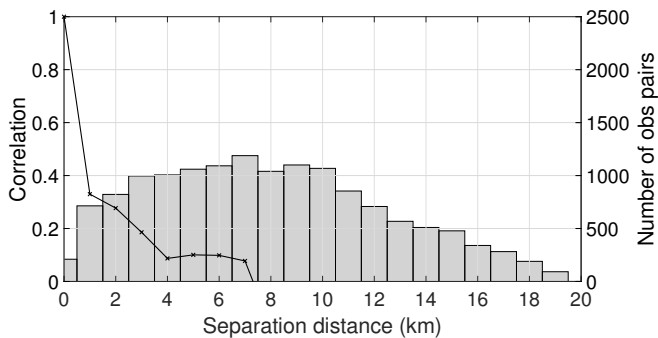

**Figure 6.** Estimated SAR WLO error correlations (black line) and number of samples (bars) used (bin width = 1km). 30th November and 1st December. Estimated error standard deviation 43cm.

## 5  Conclusions

We have shown that the Desroziers et al. (2005) diagnostic is a useful tool to identify the error covariance in WLOs from satellite

5  SAR. Further, the diagnostic has been able, in the case study, to isolate an unexpected anomaly in the correlation structure, pointing to the applicability limits of the satellite WLOs in the flood plain towards the end of the flood. The diagnostic has been useful in this study and, given its low-cost calculation, we propose it be customarily calculated in flood forecasts and hindcast analyses to support the understanding of the observation errors and to support QC protocols for selection of adequate observations. More study is needed in this context.





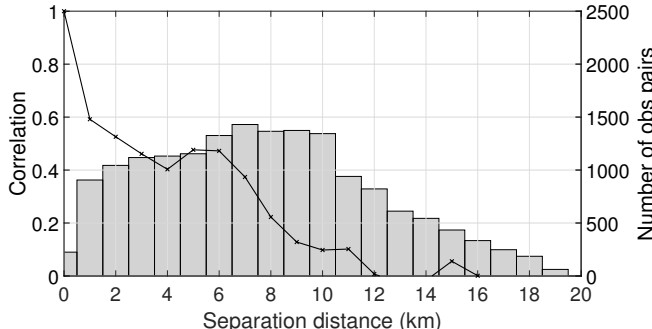

**Figure 7.** Estimated SAR WLO error correlations (black line) and number of samples (bars) used (bin width = 1km). 2nd and 4th December. Estimated error standard deviation 57cm.

*Author contributions.* JW, JG-P and DM prepared the data and ran the experiments. JW and JG-P analyzed the results and drafted the manuscript. DM, SD and NN contributed to the analysis, discussion, and manuscript editing.

*Competing interests.* The authors declare that they have no conflict of interest.

*Acknowledgements.* J. A. Waller, N. K. Nichols and S. L. Dance were supported in part by UK NERC grants NE/K008900/1 (FRANC),
5 NE/N006682/1 (OSCA). J. A. Waller and S. L. Dance received additional support from UK EPSRC grant EP/P002331/1 (DARE). N. K. Nichols was also supported by the UK NERC National Centre for Earth Observation (NCEO). J. García-Pintado, D. C. Mason and S. L. Dance were supported by UK NERC grants NE/I005242/1 (DEMON) and NE/K00896X/1 (SINATRA). The data used in this study may be obtained on request, subject to licensing conditions, by contacting the corresponding author.



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
