# Peer review of "Title: Technical note: Analysis of observation uncertainty for flood assimilation and forecasting"

_Hydrology and Earth System Sciences, 2018_

## Referee Comment (RC1) · Anonymous Referee #1 · 28 Feb 2018

The manuscript presents an analysis of the uncertainty of remote sensing-derived water level observations used for the real time constraint of flood forecast models. Specifically, the diagnostic approach presented by Desroziers et al. (2005) is applied to assess the uncertainty of spatially distributed water level observations derived from a sequence of seven high resolution SAR images acquired during the November 2012 flood event in the lower Severn and Avon rivers (UK). The hydraulic model is Lisflood-FP and the data assimilation scheme is the Local Ensemble Transform Kalman Filter. Extraction of remote sensing-derived water level values, model set-up and data assimilation scheme were presented in previous publications (respectively, Mason et al. 2012 and Garcia-Pintado et al., 2015). This manuscript firstly provides a summary of the diagnostic approach presented by Desroziers et al. (2005) and then the description of

the experimental design. In the results section, the authors provide an extensive analysis of the error statistics and quantify the spatial and temporal correlation of errors.

In my opinion this study provides an interesting contribution to the literature and it has potential application for the assimilation of remote sensing derived observations in flood forecast models. I would like to recommend the publication of this manuscript after minor revisions. Specifically, I think that the novelty of the study should be explicitly stated (the additional contribution of this study should be highlighted and explicitly compared to the previous applications of the diagnostic approach of Desroziers et al. (2005)). Moreover, I think that the overall presentation should be improved (the language is sometimes too informal), especially in the results section.

Please find my detailed comments below.

Title: I wonder whether "flood assimilation" is the most appropriate wording. The authors might consider a slight change in the title ("assimilation in flood forecast models").

Page 1, Lines 7-8 "The estimated correlations do not behave as expected": could you please be more explicit? I suggest briefly stating what the expectation was and how the results are different from the expectation. This is explained later in the manuscript (page 6), however, the abstract should provide a comprehensive (and independent) description of the topic discussed in the manuscript.

Page 2, line 13 "unbiased": is bias the only error type in Remote Sensing-derived observations of floods?

Page 2, line 18 "as is standard in other assimilation applications": I suggest rephrasing this sentence. Could you please provide examples of the "other assimilation applications"? Could you please improve the fluency of the sentence?

Page 2, line 19 "typically retaining approximately 1% of the pre-thinned observations": I suggest adding at least a reference.

Page 2, line 20 "clustered observations": the authors might consider commenting

on the algorithm presented in D.C. Mason, G.J.-P. Schumann, J.C. Neal, J. Garcia-Pintado, P.D. Bates, Automatic near real-time selection of flood water levels from high resolution Synthetic Aperture Radar images for assimilation into hydraulic models: A case study,Remote Sensing of Environment, Volume 124, 2012, Pages 705-716, ISSN 0034-4257, https://doi.org/10.1016/j.rse.2012.06.017. This algorithm was used to derive the WLOs used for the study presented in the manuscript (as stated in page 4, line 26). I think that a brief discussion on the approach and results of the algorithm (for instance, the algorithm includes the estimation of the spatial autocorrelation of the set of candidate water levels) will provide a more comprehensive analysis of the topic and enhance the impact of the study presented in this manuscript. The authors might consider adding this discussion somewhere in the manuscript (not necessarily in the introduction).

Page 2, lines 21-24: is the discussion on DA of satellite-derived soil moisture strictly relevant here?

Page 2, lines 24-30: this paragraph underlines the potential benefit of DA of RS-derived WLOs for flood forecast and the importance of a detailed understanding of observations uncertainty. I believe that moving this paragraph after line 11 could improve the readability of the manuscript. This paragraph contributes to a more general introduction on the relevance of the topic. I suggest discussing error types, data thinning, and uncertainty estimation after this general introduction.

Page 2, line 31 "directly": is this the most appropriate word? Do the authors mean "computed in a systematic way"?

Page 2, line 34 "with good results": would it be possible to clarify this statement?

Page 3, line 4: is a new paragraph required?

Page 3, line 5-6 "Thus, we then consider": could you please rephrase this sentence? (e.g. Consequently, . . .).

Page 3, line 7 "is related not related": please correct this.

Page 3, lines 8-9 "we show that …..": how is this result related to the papers cited in page 2 line 34? I would like to recommend adding a sentence to clarify the novelty of the study presented in the manuscript.

Page 3, lines 11-12: could you please improve the fluency of this sentence?

Page 3, line 15: are the superscripts correct?

Page 3, line 21: could you please rephrase the last part of the sentence?

Page 4, line 30 "error" is repeated.

Page 4, line 30 "gross error measurement": could please the authors clarify this statement?

Page 5 line 2 "typically": what do the authors mean here? "Typically" in this dataset or "typically" in the literature?

Page 5, figure 1: the authors might consider adding the underlying map (or at least the river network).

Page 6, lines 9-10: could the authors please clarify this statement?

Page 6, lines 17-18: please correct capital letters, full stop.

Page 6, line 20 "than those we estimate": I suggest rephrasing this sentence.

Page 6, line 31: the authors might consider replacing "shoulder" with something more formal (here and in the next paragraphs, e.g. page 7, line 6.

Page 7, line 6: is "are resulting in the increase…" correct? I wonder whether the authors mean that observations in different areas lead to "the increase of …"

Page 7, line 10 "and therefore the results are subject to greater sampling error": could the authors please clarify this statement?
Page 8, line 6 "we see": I suggest replacing this with something more formal.

Page 8, line 12: please rephrase this sentence.

Page 8, line 16 "to keep the forecast on track": could the authors please clarify this statement?

Page 9, line 6: the authors might consider replacing "end of the flood" with "receding limb" or something more formal than "end".

Page 9, line 9 "More study is needed in this context": the authors might want to add details on future work/research needs.

---

## Referee Comment (RC2) · Anonymous Referee #2 · 28 Feb 2018

I read the manuscript by Joanne A. Waller et al. entitled "Analysis of observation uncertainty for flood assimilation and forecasting", with a great interest.

Authors propose to assess the spatial correlation of the uncertainty of water level observations estimated via the fusion between SAR imagery-derived flood extent boundaries and a high resolution DEM. To do so, authors apply the diagnostic approach proposed by Desroziers et al. (2005) and further developed by Waller et al. (2016a) to the assimilation experiment previously proposed by Garcia-Pintado et al (2015).

I think this work is really interesting and represent a very good contribution to the research in the field of data assimilation into flood forecasting models. The manuscript is well structured and well written. The reading is smooth and easy. The scientific quality is high. However I think some discussion points are missing and some clarifi-

cations are necessary. Consequently I would suggest to consider the manuscript for publication after minor revisions.

I agree with all the comments and suggestions by reviewer 1. Especially I would suggest as well to edit the title.

Additionally, I would suggest the following edits: P 5 lines 4-6: As the standard deviation of the WLOs is estimated using rather strong hydraulic hypotheses, I would suggest to change the word "measured" by "estimated" for instance. Moreover the way the 59 cm is estimated is not so clear to me. Do you define an interpolated plane for each satellite acquisition? If not, you probably overestimate the WLO standard deviation. Could you please try to clarify this? P 4 equation 3: could you please elaborate a little more on the second term. It does not look so straightforward to me. P4 lines 30-32: I found this sentence a little difficult to read. Could you please try to clarify it? P6 lines 7-10: I think I finally understood what is done, but it took me some time. Could you please try to clarify? Especially, I would suggest to mention a "1km bin size" instead of a "1km spatial resolution" that is confusing to me. Could you please mention as well how you estimate the error spatial correlation from the covariance in the bin? Using the estimated 59 cm standard deviation of the WLOs? P6 lines 13-15: What is meant by "error of representation" is not clear to me. Could you please clarify?

Overall my little regret with the manuscript is that the observation errors are only seen from a statistical point of view. In my opinion Authors should try to elaborate a little more on how the WLOs are obtained and what could be the sources of the error spatial correlation. Authors should in my opinion refer to some interesting remarks with that respect in Mason et al 2012. Actually, the water levels used in the study are not observation from a strict point of view, but rather a piece of information derived from actual observations (the backscatter on the SAR images and the DEM). As a matter of fact each step in the process of estimating the WLOs suffer from its own sources of uncertainty. It could be worth discussing which steps of the process and which input dataset are likely to be responsible for spatially correlated errors with correlation length

up to few kilometers.

---

## Author Comment (AC1) · 26 Apr 2018

**Title**: Technical note: Analysis of observation uncertainty for flood assimilation and forecasting
**Manuscript ID**: hess-2018-43
**Authors**: J. A. Waller, J. García-Pintado, D. C. Mason, S. L Dance, N. K. Nichols

    We thank the reviewer for their positive comments, which will help improve the manuscript. Below we give each comment in bold (abridged where appropriate) and describe how we plan to alter the manuscript to address the reviewer's concern. We give suggested changes to the manuscript in italic font. We note that several of the comments ask for clarification/additional details about the derivation of the water level observations and their associated uncertainties. To address these comments we would replace Section 3 by a methodology section with the following subsections:

3.1 Derivation of water level observations.

3.2 Model and data assimilation.

3.3 Quality control and data thinning.

3.4 Potential observation error sources.

3.5 Experimental design.

The section would contain material from the current Section 3, some of the material that is currently in the introduction and include some new additional 
[revised manuscript text omitted]

---

## Author Response (AR1)

[revised manuscript text omitted]

A predominant EO technique to obtain water level observations (WLOs) is Synthetic Aperture Radar (SAR). SAR provides high-resolution observations of radar backscatter which, after processing, serve to delineate the flood extent. Then, the intersection of the flood extent with a high-resolution LiDAR Digital Terrain Model is used to obtain the WLOs. The resulting WLOs are discontinuous, but locally dense in space and consequently the errors in the observations may be highly correlated. However, the current practice when assimilating WLOs is to neglect the error correlations. To make the assumption of uncorrelated errors valid the current approach is to thin the data. Hence, in hydrology, 
[revised manuscript text omitted]

**3.1 Derivation of WLOs**

The original observations used in the deviation of WLOs are obtained using SAR which observes the surface backscatter. In a SAR image flood water appears dark so long as the surface water turbulence is insignificant. Therefore, to obtain flood extent, the pixels in a SAR image are grouped into homogeneous regions. A mean backscatter value is calculated for each region, and if this value is below a given threshold the region is classified as flooded. The threshold is determined by using training data from 'flood' and 'non-flood' regions. This initial estimate of flood extent is then refined by e.g. correcting for any high backscatter that is a result of vegetation either within the flooded region or at the flood edge; correcting for high backscatter near flooded areas that is a result of water with a rough surface; performing a 'nearest neighbour' check, where any local flood height that is significantly larger than those nearby is reclassified as non-flooded.

To provide the WLOs the refined flood extent is intersected with high resolution digital elevation model (DEM) . In order to improve the accuracy of the WLOs, they are only calculated if the slope in the DEM is sufficiently shallow. A further refinement takes into account, for example, the emergent vegetation at the flood edge.

The WLO derivation process results in a large number of WLOs that exist in clusters. It is expected that many of the observations in a cluster will be highly correlated and hence not contribute independent information. At this stage in the processing Mason et al. (2012a) thin the WLOs to reduce spatial correlation. However, we postpone this step until after the quality control procedures for the data assimilation have been performed.

**3.2 Model and data assimilation**

The observations are assimilated into a 75m resolution LISFLOOD-FP flood simulation model (Bates and Roo, 2000) using a LETKF (Hunt et al., 2007). Due to the formulation with the diagnostic described in Section 2, the localization in the LETKF is set in standard 2D Euclidean space rather than the physically based distance along the river channel described in García-Pintado et al. (2015), which would require a further adaptation of the diagnostic calculation. The localization radius is set using a compactly supported $5^{th}$ order piecewise rational function (Gaspari and Cohn, 1999, Eq. 4.10) with length scale 20km.

To compare the modeled field with the observed quantity it is necessary to define an observation operator that maps from model to observation space. In this study we use the 'nearest wet pixel' approach described in García-Pintado et al. (2013). The mapping in the nearest wet pixel approach is dependent on the inundation status at the model location. If at an observation location the model is flooded, the model equivalent of the observation is simply the water level predicted by the model. However, if the model is dry at the observation location the model equivalent of the observation is taken to be the model water level at the wet pixel nearest to the observation location.

**3.3 Quality control and data thinning**

Data assimilation techniques can lose accuracy if presented with an observation that is grossly inconsistent with the model state (Vanden-Eijnden and Weare, 2013). Thus, before being assimilated, the WLOs are subjected to several quality control (QC) protocols according to the physical characteristics of the terrain and land cover. An additional background check is performed where observations that result in anomalous observation-minus-background residuals are discarded. The QC procedures result in dense cluster of discontinuous observations in which both the observations and their errors may be highly correlated. A direct assimilation of this dense dataset would lead to an analysis biased towards the observations and, for covariance-evolving methods (e.g., ensemble Kalman filters), an over reduced posterior covariance and unstable long-term forecast/assimilation cycles. Thus, to reduce the number of correlated observations and to avoid dealing with the spatial correlation in the assimilation, the current approach is to further thin the data (as is standard in other assimilation applications such as NWP and oceanography (Dando et al., 2007; Li et al., 2010)). The applied thinning, as described in Mason et al. (2012a), uses a top down clustering approach in which principal component analysis is used to select observations that have the highest information content. The spatial autocorrelation of the resulting observations is calculated, and if any significant correlation exists the thinning procedure is applied iteratively until no significant correlation remains. Typically the thinned dataset contains approximately 1% of the

pre-thinned observations. The measured standard deviation for the thinned data set can be calculated by fitting a plane by linear regression to the WLOs. The variance of the difference between the WLO and planar surface can be used as an estimate of the observation error variance. This approach is considered adequate for this case study as the floodplain in the downstream observed areas is reasonably flat.

**3.4 Potential observation error sources**

In data assimilation the observation uncertainty has contributions from both measurement errors and representation errors. The representation error arises due to the difference between an actual observation and the modeled representation of an observation; this difference can be a result of:

– *Pre-processing/QC error:* Errors introduced during the observation pre-processing or quality control procedures.

– *Observation operator error:* Error that arises due to approximations in the mapping between model and observation space.

– *The error due to unresolved scales and processes:* Error that results from the mismatch between the scales represented in the model field and the observations.'

For the WLOs it is clear that a pre-processing error will exist as there is potential for errors to be introduced in the derivation of the WLOs. For example if the water surface is rough it may be assumed that the pixel is dry; as a result the flood extent would be incorrect and hence an error would be introduced in the WLO. For nearby pixels it is possible that there will be similar errors in the derivation process, thereby introducing correlated observation errors. The procedures in Mason et al. (2012a) provide an estimated standard deviation for the WLO pre-processing error and thin the data to ensure that the pre-processing error is uncorrelated. However, we note that in this study we use a denser dataset than is typically produced. Therefore, there is potential for some correlated pre-processing error to remain.

A potential source of correlated error for WLOs is the observation operator error. As described in Section 3.2 the observation operator uses the 'nearest wet pixel' approach. For observations in locations where the model is flooded it is expected that there is minimal error in the observation operator (since the corresponding water level is predicated directly by the model). However, if the observation location does not coincide with a flooded model pixel it is necessary to find the nearest wet pixel in the model. It is possible that in locating the nearest wet pixel and extrapolating information we introduce correlated error.

The error due to unresolved scales and processes is also a possible source of observation error correlations. Although in this case the model is of relatively high resolution compared to the observation resolution, there are still scales that are unresolved. Previous studies that have considered these scale mis-match errors have found that they are typically correlated (Janjić and Cohn, 2006; Waller et al., 2014; Hodyss and Nichols, 2015).

[Figure]

**Figure 1.** (a) Flood model domain where the colourbar denotes the height in m and (b) position of SAR WLOs on OSGB 1936 British National Grid projection; coordinates in meters. For (b) the line denotes the West/East domain split discussed in Section 4.2, Crosses: 27th, 28th 29th November, Circles: 30th November and 1st December, Squares: 2nd and 4th December.

**3.5 Calculation of WLO error statistics**

We estimate observation uncertainties for observations from a real flood event that occurred in the South West United Kingdom on an area of the lower Severn and Avon rivers in November 2012 (Figure 1a). The WLOs were extracted from a sequence of seven satellite SAR observations (acquired by the COSMO-SkyMed constellation) using the method described in Mason et al. (2012a). During the flood event the WLOs are available daily for the period $27^{th}$ November to the $4^{th}$ December 2012 (with the exception of December $3^{rd}$). Observations on the first day illustrate the flood levels just before the flood peak in the Severn. On $30^{th}$ November the river went back in bank; however, a substantial amount of water remained on the floodplain (see Fig.2 in García-Pintado et al., 2015).

Before being assimilated, the WLOs are subject to the QC and thinning procedures described in Section 3.3. 
[revised manuscript text omitted]